# Reduction in Ventilation-Induced Diaphragmatic Mitochondrial Injury through Hypoxia-Inducible Factor 1α in a Murine Endotoxemia Model

**DOI:** 10.3390/ijms23031083

**Published:** 2022-01-19

**Authors:** Li-Fu Li, Chung-Chieh Yu, Huang-Pin Wu, Chien-Ming Chu, Chih-Yu Huang, Ping-Chi Liu, Yung-Yang Liu

**Affiliations:** 1Division of Pulmonary and Critical Care Medicine, Department of Internal Medicine, Chang Gung Memorial Hospital, Keelung 20401, Taiwan; lfp3434@cgmh.org.tw (L.-F.L.); ycc@cgmh.org.tw (C.-C.Y.); whanpyng@cgmh.org.tw (H.-P.W.); rocephen@cgmh.org.tw (C.-M.C.); hcu121@cgmh.org.tw (C.-Y.H.); ewind14@cgmh.org.tw (P.-C.L.); 2Department of Internal Medicine, Chang Gung University, Taoyuan 33302, Taiwan; 3Department of Respiratory Therapy, Chang Gung Memorial Hospital, Keelung 20401, Taiwan; 4Chest Department, Taipei Veterans General Hospital, Taipei 112201, Taiwan; 5School of Medicine, Faculty of Medicine, National Yang Ming Chiao Tung University, Taipei 112304, Taiwan; 6Institute of Clinical Medicine, School of Medicine, National Yang Ming Chiao Tung University, Taipei 112304, Taiwan

**Keywords:** autophagy, biogenesis, dynamics, hypoxia-inducible factor-1α, mitochondria, ventilator-induced diaphragm dysfunction

## Abstract

Mechanical ventilation (MV) is essential for patients with sepsis-related respiratory failure but can cause ventilator-induced diaphragm dysfunction (VIDD), which involves diaphragmatic myofiber atrophy and contractile inactivity. Mitochondrial DNA, oxidative stress, mitochondrial dynamics, and biogenesis are associated with VIDD. Hypoxia-inducible factor 1α (HIF-1α) is crucial in the modulation of diaphragm immune responses. The mechanism through which HIF-1α and mitochondria affect sepsis-related diaphragm injury is unknown. We hypothesized that MV with or without endotoxin administration would aggravate diaphragmatic and mitochondrial injuries through HIF-1α. C57BL/6 mice, either wild-type or HIF-1α-deficient, were exposed to MV with or without endotoxemia for 8 h. MV with endotoxemia augmented VIDD and mitochondrial damage, which presented as increased oxidative loads, dynamin-related protein 1 level, mitochondrial DNA level, and the expressions of HIF-1α and light chain 3-II. Furthermore, disarrayed myofibrils; disorganized mitochondria; increased autophagosome numbers; and substantially decreased diaphragm contractility, electron transport chain activities, mitofusin 2, mitochondrial transcription factor A, peroxisome proliferator activated receptor-γ coactivator-1α, and prolyl hydroxylase domain 2 were observed (*p* < 0.05). Endotoxin-stimulated VIDD and mitochondrial injuries were alleviated in HIF-1α-deficient mice (*p* < 0.05). Our data revealed that endotoxin aggravated MV-induced diaphragmatic dysfunction and mitochondrial damages, partially through the HIF-1α signaling pathway.

## 1. Introduction

Mechanical ventilation (MV) is often lifesaving for patients with sepsis-related multiple organ failure. However, prolonged MV can contribute to ventilator-induced diaphragm dysfunction (VIDD), which is characterized by diaphragmatic inactivity and the rapid decline of diaphragm muscle endurance and contractility [1,2,3,4,5,6,7,8]. Sepsis-mediated profound inflammation may provoke diaphragm stretch-related injuries through the enhancement of sarcolemma membrane fragility and aggravate defects in the essential steps of the muscular respiration energy supply chain [9,10]. Furthermore, sepsis is associated with increased oxidative load, lung tissue hypoxia, and reduction in mitochondrial biogenesis of skeletal muscle [11]. Numerous studies have revealed that patients on MV commonly present with diaphragmatic weakness, which may proceed to weaning difficulty; ventilator-associated pneumonia; and worsened mortality [12,13]. Therefore, identifying the effect of the interaction between sepsis and MV on diaphragm dysfunction is clinically crucial.

Mitochondria are a predominant source of diaphragmatic free radicals—a crucial upstream mediator that initiates the signaling pathways, contributing to diaphragm muscle atrophy during endotoxemia or MV [4,14,15]. Mitochondrial DNA is an essential molecule of 16,569 bp encoding 37 genes containing unmethylated CpG in humans [16]. It modulates immune responses through messenger RNA expression induction in hypoxia-inducible factor 1α (HIF-1α) and acts as a predictor to stressors and stimuli in critically ill patients [16]. The mitochondrial DNA is stabilized through binding to mitochondrial transcription factor A (Tfam) inside the mitochondrial inner membrane [16]. Mitochondria are dynamic structures and can remodel their membranes through cycles of fission and fusion. Fission is activated through the generation of a multimeric complex comprising dynamin-related protein 1 (Drp1), and fusion is induced primarily by mitofusin (Mfn) 1 and 2 [17,18]. Fission and fusion are intimately linked to new mitochondria formation and the removal of damaged portions of mitochondria [17,18].

Mitochondria are primarily regenerated during the growth and division of pre-existing mitochondria [17]. Moreover, the activities of electron transport chain (ETC) isoform complexes II, III, and IV were observed to decline in mitochondria isolated from rat diaphragms following 12 h of MV [19]. MV-induced increase in mitochondrial free radicals is accompanied by the oxidation of lipids and proteins, leading to the derangement of mitochondrial structures [7,20]. Mitochondria with defective respiratory chain function can produce excessive electron leakage with resultant upregulation of reactive oxygen species (ROS) [7].

Sepsis may affect mitochondria through (1) increased oxidative load, accelerating lipid peroxidation and protein oxidation within mitochondria, (2) reduction in the perfusion of mitochondria, resulting in tissue hypoxia followed by cell death, (3) the inhibition of mitochondrial biogenesis, (4) increase in the opening of mitochondrial permeability transition pores, and (5) increase in autophagy [21]. Mitochondrial biogenesis is regulated mainly at the transcriptional level and requires the coordinated expression of both nuclear- and mitochondria-encoded proteins, involving peroxisome proliferator-activated receptor coactivator (PGC)-1α, Tfam, and 5ʹ-adenosine monophosphate-activated protein kinase [17,21,22].

Autophagy may protect or aggravate pulmonary diseases depending on its dichotomous functions. Basal autophagy may be required for the regulation of cell survival, whereas dysregulated autophagy may exacerbate abnormalities, including intrinsic apoptosis, mitochondrial morphological damages, and consequent muscle weakness, in the diaphragm of patients with sepsis [5,6,23].

HIF-1α is a principal regulator and can be activated and stabilized under normoxic circumstances through exposure to ROS or lipopolysaccharides (LPS), through cyclic stretch, or through contact with inflammatory cytokines [1,9,24,25]. After prolonged MV, mitochondrial ROS generation and diaphragm tissue hypoxia regulate HIF-1α expression [1,18]. Prolyl hydroxylase domain 2 (PHD2) has been the major PHD in hydroxylation modulation and the subsequent proteasomal degradation of HIF-1α [14,16,18,26]. However, the molecular mechanisms linking HIF-1α to sepsis-induced diaphragm and mitochondrial injuries remain unknown.

Using our VIDD mouse model [27], we aimed to examine the effects of MV and LPS on HIF-1α expression related to (1) the development of diaphragm and mitochondrial injuries; (2) oxidative stress, mitochondrial DNA, and downstream mitochondrial Tfam; (3) electron respiratory chains, mitochondrial dynamics, and biogenesis; (4) HIF-1α and PHD2 signaling; and (5) autophagy regulation. We hypothesized that MV with or without LPS challenge would enhance diaphragmatic mitochondrial dysfunction, generate free radicals, and cause autophagy through the activation of the HIF-1α pathway.

## 2. Results

### 2.1. Inhibition of Endotoxin-Stimulated MV-Induced Diaphragm and Mitochondrial Injuries in HIF-1α-Deficient Mice

MV of tidal volume (V_T_) 10 mL/kg with room air was administered to mice for 8 h to elicit VIDD and mitochondrial damage. The physiological conditions at the beginning and end of MV are listed in Appendix A. A stable hemodynamic status was maintained in mice through the monitoring of their mean arterial pressure. HIF-1α-deficient mice were employed to identify the role of HIF-1α in the modulation of stretch-mediated diaphragm and mitochondrial injuries following endotoxin administration. Transmission electron microscopy (TEM) was used to examine MV- and LPS-mediated alterations in diaphragm ultrastructures. Compared with mice without endotoxemia (normal saline only) subjected to V_T_ = 10 mL/kg and compared with nonventilated controls, mice with endotoxemia subjected to V_T_ = 10 mL/kg exhibited increased disarrangements in diaphragmatic myofibrillar structures with mitochondrial swelling, unclear A and I bands, tortured Z bands, and large lipid droplets (Figure 1A–D). The effects of MV on mitochondrial damage, autophagosome formation, and diaphragm function in mice with endotoxemia treated with 10 mL/kg MV were substantially attenuated in HIF-1α-deficient mice (*p* < 0.05; Figure 1E). To determine the effects of sepsis and MV on diaphragm contractile activities, we measured diaphragm muscle-specific force generation. Decreased diaphragm contractility was observed in mice with endotoxemia subjected to MV compared with those without endotoxemia subjected to MV and compared with nonventilated control mice (Figure 1F). Studies have demonstrated that the disuse atrophy of diaphragm myofibers could be a crucial contributor to weaning difficulties [9,11]. We observed substantial reduction in muscle fiber diameter in mice with endotoxemia subjected to MV compared with those without endotoxemia subjected to MV and compared with nonventilated control mice (Figure 1G). However, diaphragm weakness and atrophy were reversed in HIF-1α-deficient mice.

### 2.2. Reduction in the Effects of MV on Endotoxin-Enhanced Oxidative Stress, Mitochondrial DNA, and Mitochondrial Tfam in HIF-1α-Deficient Mice

Mitochondria are a major source of diaphragmatic ROS related to MV and sepsis-related muscle dysfunction [14]. Mitochondrial DNA is a crucial damage-associated molecular pattern (DAMP) and also a principal upstream regulator that induces the inflammatory process and cell death pathways, causing diaphragm muscle inactivity during endotoxemia or MV [15,19]. Increased levels of protein carbonyl groups and decreased production of superoxide dismutase (SOD) were observed in mice with endotoxemia subjected to MV compared with those without endotoxemia subjected to V_T_ = 10 mL/kg and compared with nonventilated control mice (Figure 2A,B). Compared with mice without endotoxemia subjected to V_T_ = 10 mL/kg and with nonventilated control mice, mice with endotoxemia subjected to MV exhibited an upregulation of mitochondrial DNA but downregulation of Tfam (a factor necessary for mitochondrial DNA replication and transcription) [16] (Figure 2C,D). Notably, the upregulation of oxidative stress and mitochondrial DNA and downregulation of SOD and Tfam were substantially reduced in HIF-1α-deficient mice.

### 2.3. Suppression of Endotoxin-Augmented MV-Induced Mitochondrial Dynamics and Biogenesis in HIF-1α-Deficient Mice

Mitochondria have the ability to remodel themselves through fusion and fission [17,18]. Western blot analyses were performed to examine the effects of MV on endotoxin-induced dynamic changes of VIDD-associated mitochondrial fission and fusion (Figure 3). Increased levels of Drp1 but decreased levels of Mfn 2 were observed in mice with endotoxemia subjected to V_T_ = 10 mL/kg compared with those without endotoxemia subjected to V_T_ = 10 mL/kg and compared with nonventilated control mice. The enhanced expression of Drp1 and suppressed expression of Mfn2 caused by endotoxemia and MV at V_T_ = 10 mL/kg were substantially reversed in HIF-1α-deficient mice. The mitochondrial organelles are replaced through biogenesis and are transcriptionally regulated through Tfam, PGC-1α, and functional respiratory chain enzymes [7,17,19]. The downregulation of succinate dehydrogenase (SDH, complex II), cytochrome-c oxidase (complex IV), mitochondrial cytochrome C (a marker of mitochondrial structure integrity), and PGC-1α (a regulator of muscle oxidative capacity) were observed in mice with endotoxemia subjected to V_T_ = 10 mL/kg in contrast to those without endotoxemia subjected to V_T_ = 10 mL/kg and nonventilated control mice (Figure 4). Reduced mitochondrial biogenesis after MV was substantially mitigated in HIF-1α-deficient mice (Figure 4).

### 2.4. Inhibition of the Effects of MV on Endotoxin-Induced HIF-1α and PHD2 Protein Expression in HIF-1α-Deficient Mice

An increase in mitochondrial ROS regulates HIF-1α activation [14]. PHD2 is a negative regulator of HIF-1α [26]. Western blot analyses revealed upregulated HIF-1α but downregulated PHD2 in mice with endotoxemia subjected to V_T_ = 10 mL/kg in contrast to those without endotoxemia subjected to V_T_ = 10 mL/kg and nonventilated control mice (Figure 5). However, these characteristics were reversed in HIF-1α-deficient mice.

### 2.5. Reduction in the Effects of MV on Endotoxin-Enhanced VIDD and Autophagy in HIF-1α-Deficient Mice

Autophagy is an essential process characterized by the removal of injured cellular components, such as mitochondria, in skeletal muscles, and it is designed to maintain the quality of mitochondrial networks [4,5,17]. The effects of MV on different parameters, including an increase in mitochondrial injury, autophagosomes, and light chain 3-II (LC3-II) expression (a marker of increased autophagosome production) and a decrease in P62 expression (a marker of increased autophagic flux), in mice with endotoxemia treated with V_T_ = 10 mL/kg were substantially restored in HIF-1α-deficient mice (Figure 6). Our results jointly suggest that endotoxin-induced and concurrent MV-induced oxidative stress impair mitochondrial dynamics and that biogenesis in the diaphragm is ameliorated through the inhibition of the HIF-1α pathway (Figure 7).

## 3. Discussion

In the present study, we demonstrated that (1) LPS can augment MV-induced mitochondrial oxidative stress and reduce antioxidant activity; (2) LPS can increase MV-induced mitochondrial DNA generation and decrease Tfam expression; (3) LPS can upregulate MV-induced mitochondrial fission protein Drp1 and downregulate mitochondrial fusion protein Mfn2; (4) LPS can reduce the activity of mitochondrial respiratory chain complexes II, III, and IV and the levels of PGC-1α during MV; (5) LPS can upregulate MV-induced HIF-1α expression and downregulate PHD2; (6) LPS can increase MV-induced autophagic markers LC3-II and decrease the level of the autophagic degradation indicator P62; and (7) LPS can aggravate MV-induced diaphragmatic ultrastructural damage and reduce the cross-sectional area and contractility of injured diaphragm myofibers. However, these harmful effects on LPS-stimulated VIDD associated with mitochondrial dysfunction were attenuated in mice with the HIF-1α gene knocked out. Sepsis is a severe overwhelming systemic inflammation caused by a dysregulated host immune response to infection. Despite the availability of advanced medical therapy for sepsis, mortality due to multiorgan failure remains high [28]. MV is indispensable for adequate oxygenation and ventilation in patients with sepsis. MV with even optimal V_T_ synergistically augments lung injury in sepsis [29]. Sepsis can sensitize the sarcolemma of myofibrils to stress-related diaphragm injury [30]. A recent study reported that patients with sepsis under MV develop earlier and more intense diaphragm dysfunction than those without sepsis [31]. The severe destruction of mitochondrial ETC in skeletal muscle was observed in the biopsies of patients with septic shock within the first 24 h after intensive care unit (ICU) admission [32]. Brealey et al. reported a considerable correlation between muscle mitochondrial dysfunction and mortality in sepsis [11]. Accumulating evidence suggests that MV and sepsis exert synergistic effects on the development of severe diaphragm weakness and ventilator-associated complications and dependence [30,33]. The underlying pathophysiological mechanism is complicated, involving increased protein degradation, decreased protein synthesis, and mitochondrial dysfunction, leading to bioenergetic failure [7,18,34].

The principal feature of sepsis is tissue hypoxia caused by defects in macrocirculation and microcirculation and inflammation-associated hypoxia [35]. Mitochondria are the primary source of cellular energy—regulating innate and adaptive immunity and cell death. At the cellular milieu, hypoxia correlates with altered oxidative phosphorylation, increased oxygen consumption, and cellular energetic imbalance due to deteriorated activity of ETC complexes I–V [18]. Sepsis-related inflammatory hypoxia results in the massive production of mitochondrial ROS, and an insufficient scavenging of the ROS may damage the mitochondrial structure and impair the DNA, dynamics, and biogenesis units of intact mitochondria [35]. The mitochondrial genome appears to be susceptible to ROS-evoked DNA damage, and mitochondrial DNA mutations leading to mitochondrial respiratory chain dysfunction contribute to reducing muscle endurance in humans [36]. The destruction of mitochondrial membrane integrity or destabilization of mitochondrial DNA packaging caused by Tfam deficiency may release mitochondrial DNA into systemic circulation, and these mitochondrial DNA fragments, behaving as DAMPs, interact with Toll-like receptor 9 or inflammasomes in sepsis [16]. Mitochondrial DNA has the capacity to induce and propagate inflammatory reactions in sepsis and is a useful predictor for ICU-related mortality [37]. During sepsis, oxidative stress impairs mitochondrial ETC subunits, leading to insufficient adenosine triphosphate (ATP) production and a decline in muscle endurance, and also provokes proteolytic enzyme pathways, contributing to a decrease in contractile proteins and muscle strength [38]. The mechanisms responsible for mitochondrial dysfunction during sepsis are the suppression of pyruvate in the citric acid cycle, inhibition of the respiratory chain complex, reduction in the use of oxidative phosphorylation substrate, presence of oxidative stress-induced membrane damage, and decrease in mitochondrial constituents [39]. In the present study, we demonstrated that exaggerated mitochondrial ROS and reduced SOD activity occur along with mitochondrial DNA upregulation and Tfam deficiency in our VIDD with endotoxemia model.

HIF-1α is a pivotal nuclear transcription factor that regulates cellular oxygen homeostasis to hypoxia. Furthermore, HIF-1α can be stabilized under normoxic conditions during inflammation, and this activation is associated with drastic levels of proinflammatory cytokines and a decrease in cellular oxygen consumption [39]. LPS, cyclic stretching due to MV, or mitochondrial DNA can block SDH activity and facilitate succinate production, resulting in the inhibition of PHD2, suppression of both polyubiquitination and proteasomal degradation, and promotion of HIF-1α stabilization in vitro and in vivo [1,3,13,16,40]. Peyssonnaux et al. reported that HIF-1α activates numerous pivotal proinflammatory cytokines participating in the pathogenesis of LPS-induced sepsis and that HIF-1α inhibition is a defense mechanism against LPS-induced sepsis and mortality [41]. Mounting evidence suggests the irreplaceable role of HIF-1α in the regulation of mitochondrial homeostasis [39]. HIF-1α activates mitophagy, inhibits mitochondrial oxidative capacity, and negatively regulates mitochondrial respiration, which are bioenergetic steps necessary for adaption to hypoxia [42]. Although mitochondrial dysfunction is demonstrated to be involved in the pathophysiology of diaphragm injury in critical illness, the interplay between HIF-1α and mitochondria during MV or endotoxemia is not well known. To our knowledge, this is the first study to investigate the role of HIF-1α in the regulation of mitochondrial dysfunction. Tissue hypoxia during sepsis triggers HIF-1α activation for adaption to hypoxia, but overexpression of HIF-1α, in turn, leads to mitochondrial dysfunction.

Picard et al. posited that the potential mechanisms of mitochondrial abnormalities in MV diaphragms are the downregulation of numerous genes encoding proteins to facilitate mitochondrial DNA–related mitochondrial biogenesis and alteration of mitochondrial fission–fusion dynamics [7]. The promotion of mitochondrial fission protein Drp1 plays a crucial role in mitochondrial fission activation, but Drp1 overexpression results in mitochondrial fragmentation, depolarization of the transmembrane potential, and promotion of mitochondrial ROS generation. ROS can in turn profoundly activate Drp1 formation to accelerate mitochondrial fission frequency, leading to mitochondrial dysfunction [18]. Conversely, mitochondrial fusion is imperative for mitochondrial DNA stability in skeletal muscles and protection against mitochondrial DNA mutations [43]. Under physiological circumstances, mitochondria continue to undergo a dynamic balance of fission and fusion [44,45], whereas the balance is interrupted during sepsis, which presents as drastic mitochondrial fission and deficient mitochondrial fusion, along with a decrease in mitochondrial quality [44]. The dysregulation of mitochondrial dynamics could result in ROS production and mitochondrial DNA outflow, which worsen mitochondrial bioenergetic efficiency, contributing to cellular dysfunction in sepsis [45]. The imbalance of mitochondrial dynamics and subsequent mitochondrial dysfunction can facilitate inappropriate apoptotic signaling and lead to diaphragm atrophy [15]. In the present study, HIF-1α gene knockout improved the mitochondrial dynamic balance through Mfn2 upregulation to promote mitochondrial fusion and Drp1 downregulation to inhibit mitochondrial fission in our LPS-stimulated VIDD model.

PGC-1α is identified as the transcriptional coactivator of mitochondrial function and can regulate mitochondrial biogenesis (e.g., mitochondrial DNA replication and transcription and mitochondrial respiration) in skeletal muscles. PGC-1α activates the downstream bioenergetic target, such as mitochondrial Tfam, a gene essential for mitochondrial biogenesis, which reflects the alterations of mitochondrial DNA in the cells and plays a pivotal role in mitochondrial DNA maintenance [18]. However, PGC-1α action can be downregulated by systemic inflammatory reactions through nuclear factor-κB signaling in the peripheral skeletal muscle [46]. Oliveira et al. reported that the diminished expression of detoxifying ROS enzymes (e.g., SOD 2) and decreased mitochondrial energy production contribute to a significant decrease in PGC-1α expression in the septic diaphragm [46]. Picard et al. demonstrated that VIDD is associated with mitochondrial dysfunction, as indicated by a decreased activity of respiratory chain enzymes SDH and cytochrome-c oxidase, which are vulnerable to mitochondrial dysfunction in injured human diaphragms [7]. These observations agree with our results that the antioxidant ability and mitochondrial ETC complexes II, III, and IV activities were reduced in our LPS-stimulated VIDD animal model. Sepsis increases mitochondrial ROS, elicits ETC abnormalities, and decreases ATP production by mitochondria in multiorgan dysfunction [18]. Mitochondrial biogenesis is accompanied by an increased antioxidant defensive response to oxidative stress by PGC-1 transcriptional coactivators [47]. Mitochondria with reduced respiratory ETC function can enable drastic electron leakage with a subsequent upregulation in ROS production [48]. Oxidative stress has the capacity to suppress contractile ability directly and is also crucial in the promotion of several proteolytic pathways involved in muscle atrophy [7]. In the present study, HIF-1α gene knockout improved the activity of ETC complexes II–IV and mitochondrial biogenesis and restored the antioxidative ability of mitochondria through PGC-1α upregulation in our LPS-stimulated VIDD model.

Autophagy, a cytoprotective process, is required for the metabolism and turnover of damaged organelles [8]. Mitophagy is a form of selective autophagy and is activated to both eliminate dysfunctional mitochondria and preserve mitochondrial quality control. Mitochondrial protein ubiquitination mediates the degradation of injured mitochondria through mitophagy. Mitochondrial ubiquitination is enhanced by Sequestosome 1/p62 during mitophagy [49]. ROS can promote the autophagy–lysosomal pathway, resulting in protein degradation implicated in skeletal muscle proteolysis and diaphragm atrophy in VIDD. Mitochondria are insulated by double-membraned vesicles called autophagosomes, which are regulated by LC3 during mitophagy. The assessment of the conversion of LC3-1 to LC3-II is known as an autophagic marker [50]. In the present study, we determined that MV and LPS can further induce autophagy, as verified by increased autophagosomes in proximity to mitochondria based on the TEM analysis and enhanced expression of LC3-II and declined levels of p62. Notably, dysregulated autophagy and subsequent autophagosome formation associated with mitochondrial dysfunction during MV and endotoxemia are mitigated by HIF-1α gene knockout.

In an in vitro study of human umbilical vein endothelial cells, Dewangan et al. demonstrated that salinomycin, a monocarboxylic polyether ionophore obtained from Streptomyces albus, reduced angiogenesis of breast cancer cell progression through inhibiting the HIF-1α/vascular endothelial growth factor signaling axis [51]. Hu et al. also reported that chetomin, a small molecule inhibitor of HIF-1α, repressed transcriptional activities of HIF-1α and its downstream target lactate dehydrogenase A [52]. This in turn resulted in an appropriate metabolic shift from aerobic glycolysis to oxidative phosphorylation and was accompanied by an increase in mitochondrial content and cellular ATP levels. Further investigation using these inhibitors is necessary to prove their effects in ameliorating VIDD. In conclusion, we demonstrated that MV with endotoxemia can augment mitochondrial dysfunction, determined based on transcription regulation, mitochondrial dynamics, biogenesis, mitochondrial autophagy (mitophagy), and the ultrastructural and functional impairment of the diaphragm through HIF-1α signaling pathways by using our established animal model, which accords closely with real clinical situations. Findings on the relationship between HIF-1α and mitochondrial dysfunction during MV- and sepsis-related diaphragm dysfunction may reveal how clinicians can best manipulate mitochondrial function enhancement through applicable HIF-1α inhibitors [51,52] to improve sepsis-related diaphragm dysfunction in patients undergoing MV.

## 4. Materials and Methods

### 4.1. Experimental Animals

Wild-type or HIF-1α-deficient C57BL/6 mice, weighing between 20 and 25 g, aged between 6 and 8 weeks, were obtained from Jackson Laboratories (catalog number 007227, Bar Harbor, ME, USA) and National Laboratory Animal Center (Taipei, Taiwan) [53]. The HIF-1αfl/fl, mouse line was bred to C57BL/6 mice carrying a CD4cre transgene. In the resulting offspring, a region encompassing exon 2 was excised in CD^4+^ cells. Genotyping on tail DNA was performed as previously described [53]. An amplification of ~250 bp by primers DP11 (5′-GCAGTTAAGAGCACTAGTTG) and DP12(5′-GGAGCTATCTCTCTAGACC) indicated the presence of a floxed HIF-1α allele. An amplification of ~200 bp indicated a wild-type HIF-1α allele. PCR was used to genotype tail DNA for the presence of CD4cre (forward 5′-CGATGCAACGAGTGATGAGG, reverse 5′-CGCATAACCAGTGAAACAGC). CD4-Cre is the promoter element and HIF-1α is only knocked down in CD^4+^ cells [53]. The study was performed in strict accordance with the recommendations in the Guide for the Care and Use of Laboratory Animals of the National Institutes of Health (NIH). The protocol was approved by the Institutional Animal Care and Use Committee of Chang Gung Memorial Hospital (Permit number: 2017111001). All surgery was performed under xylazine and Zoletil anesthesia, and all efforts were made to minimize suffering.

### 4.2. Experimental Groups

Animals were randomly distributed into 5 groups in each experiment: group 1, nonventilated control wild-type mice with normal saline; group 2, nonventilated control wild-type mice with LPS; group 3, V_T_ 10 mL/kg wild-type mice with normal saline; group 4, V_T_ 10 mL/kg wild-type mice with LPS; group 5, V_T_ 10 mL/kg HIF-1α^−/−^ mice with LPS; In each group, three mice underwent TEM, and five mice underwent measurement for immunohistochemistry assay, ROS, and western blots.

### 4.3. Relative Mitochondrial and Nuclear DNA Ratio

The mitochondrial DNA of murine diaphragm was measured by real-time PCR on total DNA extracted using the DNeasy Blood and Tissue Kit (Qiagen Ltd., Manchester, UK). Primer sequences for the mitochondrial segment were as follows: forward primer CCGCAAGGGAAAGATGAAAGAC and reverse primer TCGTTTGGTTTCGGGGTTTC. Primer sequences for the single-copy nuclear control were as follows: forward primer GCCAGCCTCTCCTGATTTTAGTGT and reverse primer GGGAACACAAAAGACCTCTTCTGG [54]. All quantity PCR reactions using SYBR Master Mix were performed on an ABI Prism 7000 sequence detector PCR system (Applied Biosystems, Foster City, CA, USA). Mitochondrial DNA was calculated relative to nuclear DNA using the following equations: Δ*C*_T_ = mitochondrial *C*_T_ − nuclear *C*_T_ and relative mitochondrial DNA content = 2 × 2^−ΔCT^.

### 4.4. Analysis of Electron Transport Chain Complex Activity

The activities of ETC were detected in mouse diaphragm using complex I to IV colorimetric microplate assay kit (NADH dehydrogenase (complex I), succinate dehydrogenase (SDH, complex II), ubiquinol cytochrome C oxidoreductases (complex III), and cytochrome c oxidase (COX, complex IV)) according to the manufacturer’s instructions (Abcam, Cambridge, UK). The enzymatic activities of mitochondrial electron transport chain are expressed as the changes in absorbance per minutes per microgram protein.

### 4.5. Cross-Sectional Area of Muscle Fibers

The diaphragms were paraffin-embedded, sliced at 4 μm, and stained with hematoxylin and eosin (H&E). The cross-sectional areas, a semi-quantitative method, were reviewed from 50 muscle fibers by a single investigator blinded to therapeutic category of the mouse and analyzed using NIH image 1.6 software.

### 4.6. Measurement of Diaphragm Force-Frequency Relationships

Diaphragm-specific force generation was assessed as previously reported [50]. Briefly, the diaphragm strips, 3–5 mm in diameter and 1.5 cm in length, were dissected from the left costal diaphragm and mounted vertically in tissue baths containing 6 mL Krebs-Hensleit solution of the following composition: NaCl 135 mM, KCl 5 mM, dextrose 11.1 mM, CaCl_2_ 2.5 mM, MgSO_4_ 1 mM, NaHCO_3_ 14.9 mM, NaHPO_4_ 1 mM, insulin 50 units/L, d-tubocurarine 12 μM). The tissue bath solution was maintained at 25 °C and gassed with 95% O_2_ plus 5% CO_2_ (pH 7.4). The rib end of the strips was attached to the bottom of the baths by silk ties, and the tendon end was tied by a fine silk thread and connected to a force transducer (FORT25, WPI, Sarasota, FL, USA). External stimuli were delivered by an electronic stimulator (S88; Grass Instrument Division, Astro-Med, Inc., West Warwick, RI, USA) via bipolar platinum electrodes. Diaphragmatic contractions were recorded isometrically by a data acquisition system (PowerLab /4sp; ADInstruments, Pty Ltd., Castle Hill, Australia) fitted with a bridge amplifier (QUAD bridge amplifier; ADInstruments) via Chart software (version 5.0; AD Instruments) for off-line analysis. After a 15 min equilibrium period, muscle length was adjusted to L_o_ (the length at which force generation was maximum) and stimulation current was adjusted to supramaximal levels and a force–frequency curve was constructed by stimulating strips at 15, 30, 60, 100, and 160 Hz (train duration 500 msec) with a 10 s rest period between adjacent stimulus trains. After raw force measurements were completed, diaphragmatic contractility was normalized as specific force. Cross-sectional area was calculated as muscle strip weight divided by muscle density (1.06) and muscle length; diaphragm specific force generation was then calculated as raw force divided by cross-sectional area.

### 4.7. Immunoblot Analysis

The diaphragms were homogenized in 0.5 mL of lysis buffer, as previously described [27]. Crude cell lysates were matched for protein concentration, resolved on a 10% bis-acrylamide gel, and electrotransferred to Immobilon-P membranes (Millipore Corp., Bedford, MA, USA). For assay of Drp1, HIF-1α, LC3-II, Mfn2, P62, PHD2, Tfam, and GAPDH, Western blot analyses were performed with respective antibodies (New England BioLabs, Beverly, MA, USA and Santa Cruz Biotechnology, Santa Cruz, CA, USA,). Blots were developed by enhanced chemiluminescence (NEN Life Science Products, Boston, MA, USA).

### 4.8. Analysis of Data

The Western blots were quantitated using an NIH image analyzer Image J 1.27z (National Institutes of Health, Bethesda, MD, USA) and presented as arbitrary units. Values were expressed as the mean ± SD from at least 5 separate experiments. The data of protein oxidation, histopathologic assay, and oxygenation were analyzed using Statview 5.0 (Abascus Concepts, Cary, NC, USA; SAS Institute). All results of real-time PCR and Western blots were normalized to the nonventilated control wild-type mice with LPS. ANOVA was used to assess the statistical significance of the differences, followed by multiple comparisons with a Scheffe’s test, and a *p* value < 0.05 was considered statistically significant. Additional details, including lipopolysaccharide administration, measurement of diaphragmatic oxidative stress and antioxidant enzyme expression, mitochondrial injury score, mitochondrial isolation, TEM, and ventilator protocol were performed as previously described [27,50].

## Figures and Tables

**Figure 1 ijms-23-01083-f001:**
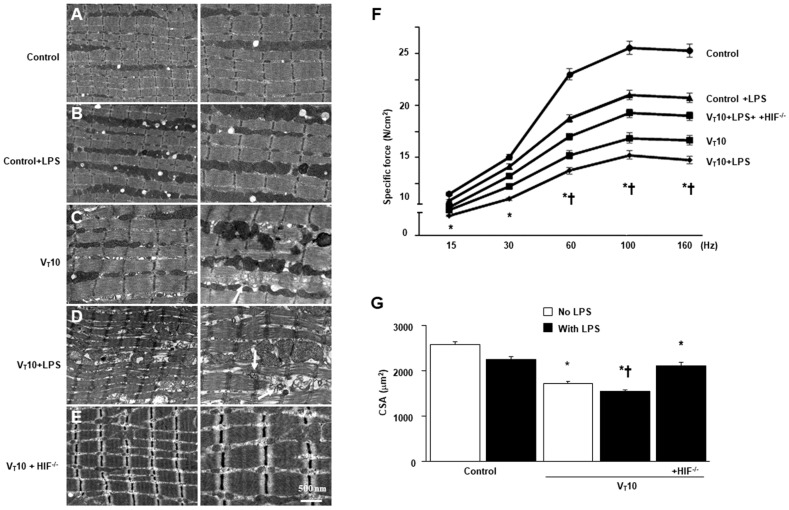
Suppression of endotoxin-enhanced mechanical ventilation-induced diaphragm and mitochondrial injuries in HIF-1α-deficient mice. Representative micrographs of the longitudinal sections of diaphragm (×20,000: left panel; ×40,000: right panel) were from the same diaphragms of nonventilated control mice and mice ventilated at a tidal volume (V_T_) of 10 mL/kg (V_T_ 10) for 8 h (*n* = 3 per group). (**A**,**B**) Nonventilated control wild-type mice with or without LPS treatment: normal sarcomeres with clear A bands, I bands, and Z bands; (**C**) 10 mL/kg wild-type mice without LPS treatment (normal saline): increase of diaphragmatic disarray; (**D**) 10 mL/kg wild-type mice with LPS treatment: disruption of sarcomeric structure, mitochondrial swelling with a vacuole-like structure, streaming of Z bands, and collection of lipid droplets; (**E**) 10 mL/kg HIF-1α deficient mice: reduction of diaphragmatic disruption. (**F**) Diaphragm muscle-specific force production was measured as described in Methods. (**G**) Cross-sectional area of diaphragm muscle fiber was measured as described in Methods (*n* = 5 per group). Mitochondrial swelling with concurrent formation of vacuoles and autophagosomes containing heterogeneous cargo are identified by arrows. * *p* < 0.05 versus the nonventilated control mice with LPS treatment; † *p* < 0.05 versus all other groups. Scale bar represents 500 nm. CSA = cross-sectional area; HIF^−/−^ = hypoxia-inducible factor-1α-deficient mice; Hz = hertz; LPS = lipopolysaccharide; N = newton.

**Figure 2 ijms-23-01083-f002:**
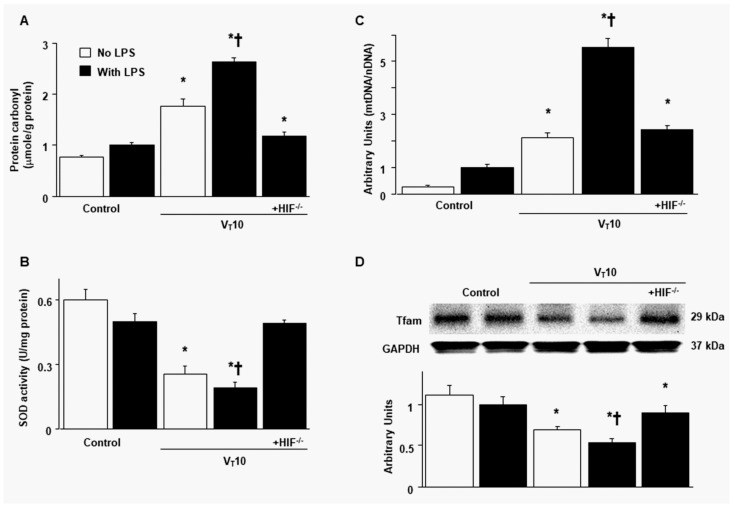
Inhibition of endotoxin-augmented mechanical ventilation-induced oxidative load, mitochondrial DNA, and mitochondrial transcription factor A in HIF-1α-deficient mice. (**A**) Protein carbonyl groups and (**B**) SOD were from the diaphragms of nonventilated control mice and mice ventilated at a tidal volume of 10 mL/kg for 8 h with or without LPS administration (*n* = 5 per group). (**C**) Real-time PCR performed for mitochondrial DNA expression was from the diaphragms of nonventilated control mice and mice ventilated at a tidal volume of 10 mL/kg for 8 h with or without LPS administration (*n* = 5 per group). Arbitrary units were expressed as the ratio of mitochondrial DNA to nuclear DNA (*n* = 5 per group). (**D**) Western blots were performed using antibodies that recognize mitochondrial transcription factor A and GAPDH expression from the diaphragms of nonventilated control mice and mice ventilated at a tidal volume of 10 mL/kg for 8 h with or without LPS administration (*n* = 5 per group). Arbitrary units were expressed as relative Tfam activation (*n* = 5 per group). * *p* < 0.05 versus the nonventilated control mice with LPS treatment; **†**
*p* < 0.05 versus all other groups. GAPDH = glyceraldehydes-phosphate dehydrogenase; mt = mitochondria; nDNA = nuclear DNA; SOD = superoxide dismutase; Tfam = mitochondrial transcription factor A.

**Figure 3 ijms-23-01083-f003:**
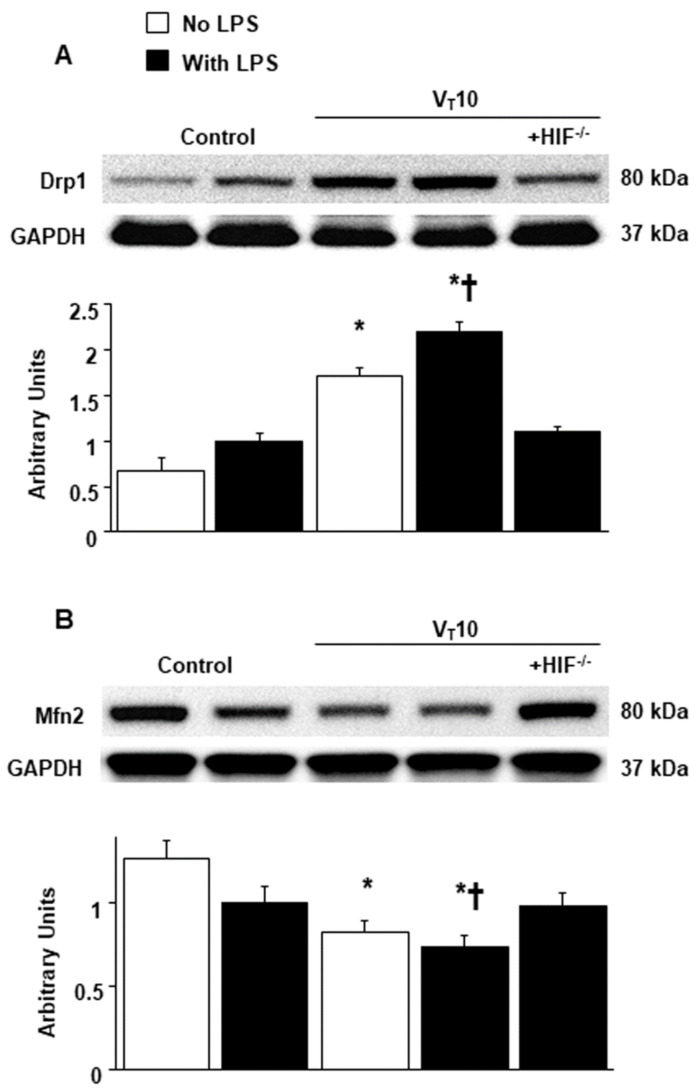
Reduction of endotoxin-aggravated mechanical ventilation-mediated mitochondrial dynamics in HIF-1α-deficient mice. Western blots were performed using antibodies that recognize Drp1 (**A**), Mfn2 (**B**), and GAPDH expression from the diaphragms of nonventilated control mice and mice ventilated at a tidal volume of 10 mL/kg for 8 h with or without LPS administration (*n* = 5 per group). Arbitrary units were expressed as relative Drp1 and Mfn2 activation (*n* = 5 per group). * *p* < 0.05 versus the nonventilated control mice with LPS treatment; **†**
*p* < 0.05 versus all other groups. Drp1 = dynamin-related protein 1; Mfn2 = mitofusion 2.

**Figure 4 ijms-23-01083-f004:**
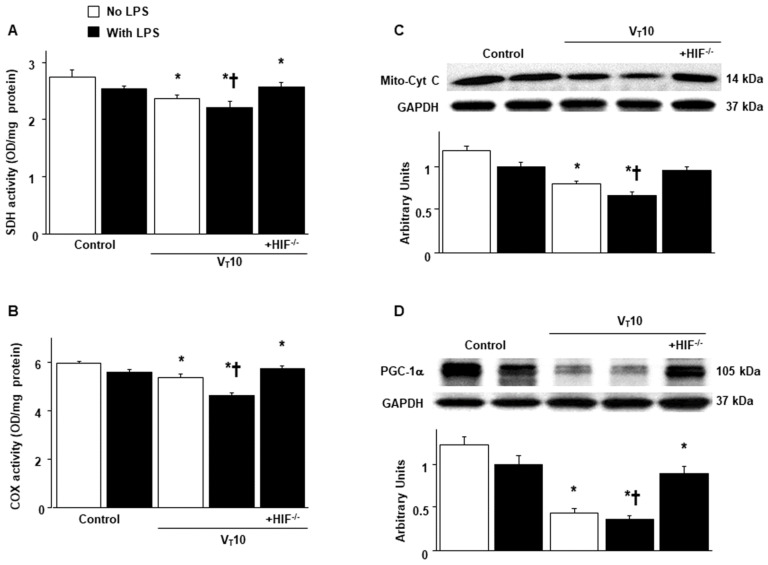
Abrogation of endotoxin-enhanced mechanical ventilation-mediated mitochondrial biogenesis in HIF-1α-deficient mice. (**A**) SDH (complex II)) and (**B**) COX (complex IV) were from the diaphragms of nonventilated control mice and mice ventilated at a tidal volume of 10 mL/kg for 8 h with or without LPS administration (*n* = 5 per group). Western blots were performed using antibodies that recognize mitochondrial cytochrome C (**C**), PGC-1α (**D**), and GAPDH expression from the diaphragms of nonventilated control mice and mice ventilated at a tidal volume of 10 mL/kg for 8 h with or without LPS administration (*n* = 5 per group). Arbitrary units were expressed as relative mitochondrial cytochrome C and PGC-1α activation (*n* = 5 per group). * *p* < 0.05 versus the nonventilated control mice with LPS treatment; **†**
*p* < 0.05 versus all other groups. COX = cytochrome-c oxidase; Mito-Cyt C = mitochondrial cytochrome C; PGC-1α = peroxisome proliferator activated receptor-γ coactivator; SDH = succinate dehydrogenase.

**Figure 5 ijms-23-01083-f005:**
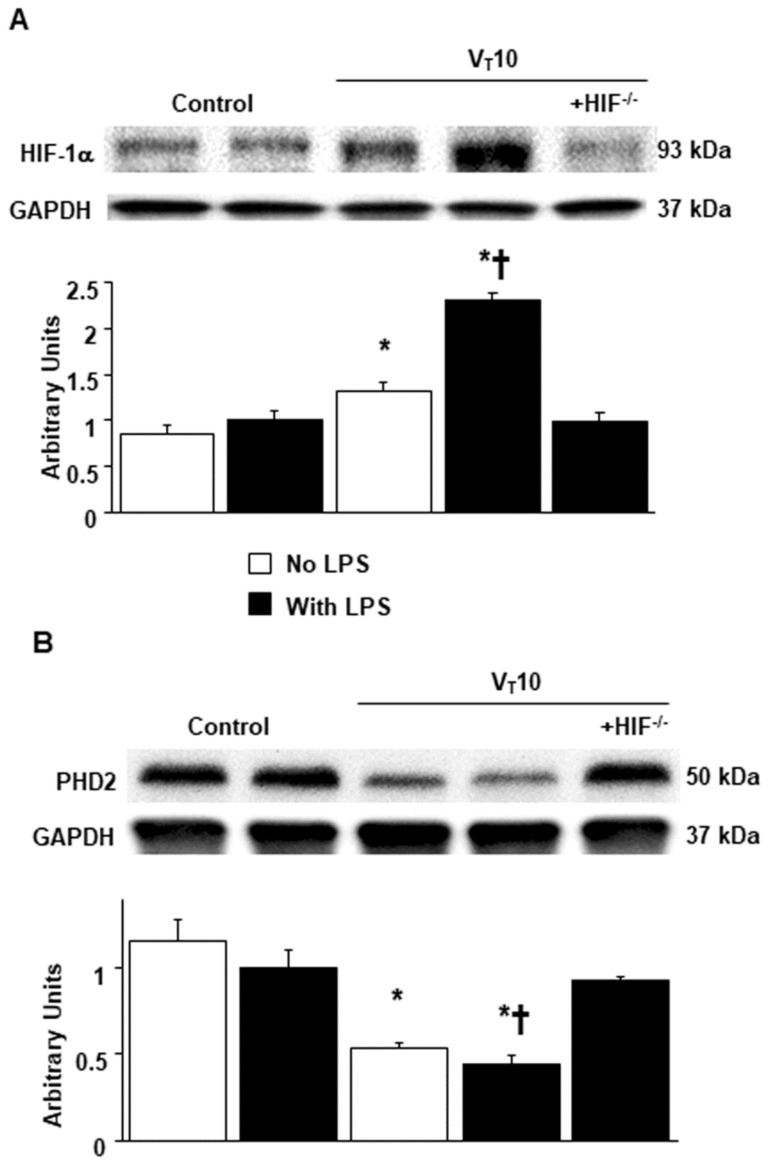
Amelioration of endotoxin-augmented mechanical ventilation-mediated HIF-1α and PHD2 protein expression in HIF-1α-deficient mice. Western blots were performed using antibodies that recognize HIF-1α (**A**), PHD2 (**B**), and GAPDH expression from the diaphragms of nonventilated control mice and mice ventilated at a tidal volume of 10 mL/kg for 8 h with or without LPS administration (*n* = 5 per group). Arbitrary units were expressed as relative HIF-1α and PHD2 activation (*n* = 5 per group). * *p* < 0.05 versus the nonventilated control mice with LPS treatment; **†**
*p* < 0.05 versus all other groups. PHD2 = prolyl hydroxylase domain 2.

**Figure 6 ijms-23-01083-f006:**
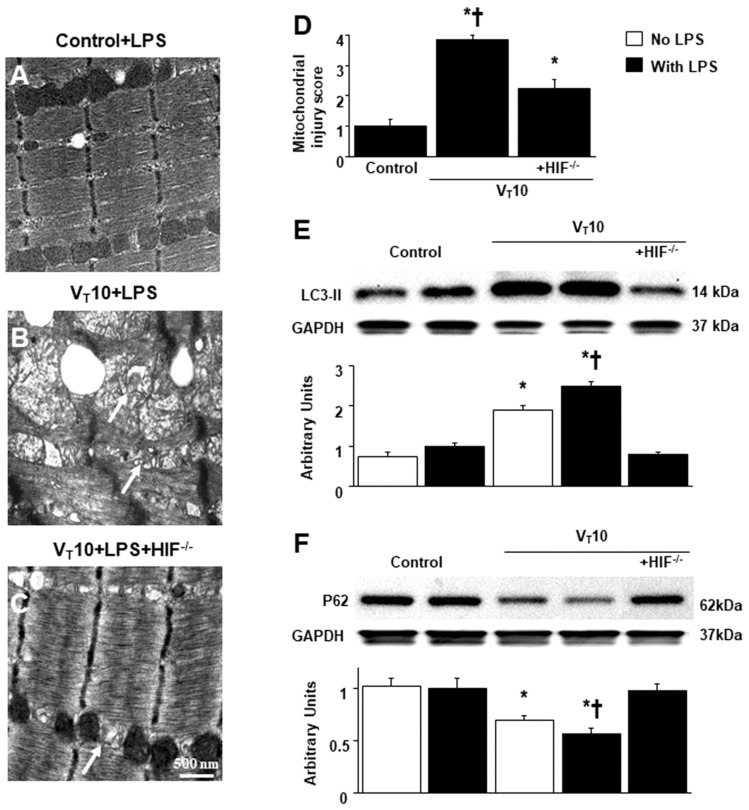
Inhibition of endotoxin-aggravated mechanical ventilation-induced diaphragm injury and autophagy in HIF-1α deficient mice. (**A**–**D**) Representative micrographs of the longitudinal sections of diaphragm (×40,000) were from the diaphragms of nonventilated control mice and mice ventilated at a tidal volume of 10 mL/kg for 8 h with LPS administration (*n* = 3 per group). Mitochondrial damage with coexisting autophagosomes containing heterogeneous cargo, loss of cristae, and vacuole formation is identified by arrows. Western blots were performed using antibodies that recognize LC3-II (**E**), P62 (**F**), and GAPDH expression from the diaphragms of nonventilated control mice and mice ventilated at a tidal volume of 10 mL/kg for 8 h with or without LPS administration (*n* = 5 per group). Arbitrary units were expressed as relative LC3-II and P62 activation (*n* = 5 per group). * *p* < 0.05 versus the nonventilated control mice with LPS treatment; **†**
*p* < 0.05 versus all other groups. LC3-II = light chain 3-II.

**Figure 7 ijms-23-01083-f007:**
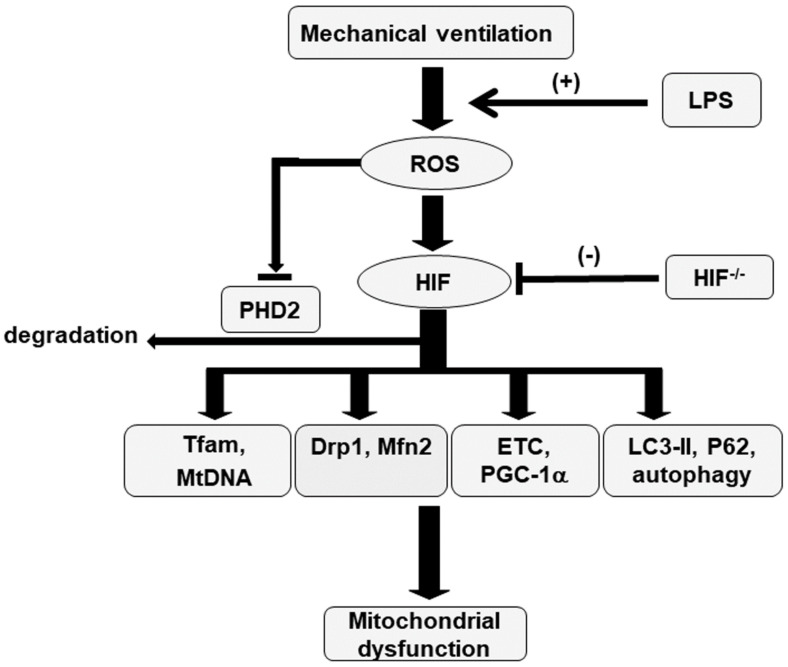
Schematic figure illustrating the signaling pathway activation with mechanical ventilation and endotoxemia. Endotoxin-induced augmentation of mechanical stretch-mediated oxidative stress, mitochondrial dynamics and biogenesis, and autophagy, and diaphragm damage were alleviated with HIF-1α homozygous knockout. Drp1 = dynamin-related protein 1; ETC = electron transport chain; HIF = hypoxia-inducible factor; LC3-II = light chain 3-II; LPS = lipopolysaccharide; Mfn2 = mitofusion 2; Mt = mitochondria; PGC-1α = peroxisome proliferator activated receptor-γ coactivator-1α; PHD2 = prolyl hydroxylase domain 2; ROS = reactive oxygen species; Tfam = mitochondrial transcription factor A; VIDD = ventilator-induced diaphragm dysfunction.

## Data Availability

The data presented in this study are available on request from the corresponding author.

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
