# Peer review of "Reduction in Ventilation-Induced Diaphragmatic Mitochondrial Injury through Hypoxia-Inducible Factor 1α in a Murine Endotoxemia Model"

_ijms, 2022, doi:10.3390/ijms23031083_

Round 1

Reviewer 1 Report

Thanks for allowing me to review this interesting paper.

In their work Li-Fu Li and colleagues address a relevant problem in the icu, ventilation-induced diaphragm dysfunction (VIDD).
In their work they show that the detrimental effect of mechanical ventilation on diaphragmatic activity is markedly enhanced in case of endotoxemia; however, this effect is reduced in HIF-1α-deficient mice.

The authors focused on the mithocondrial pathways analyzing 5 experimental groups: 1 non ventilated control wild-type mice with normal saline; 2 non ventilated control wild-type mice with LPS; 3 ventilated wild-type mice with normal saline; 4 ventilated wild-type mice with LPS; 5, ventilated HIF-1α-deficient mice with LPS.

Summarizing, they found that:

  1. Endotoxemia + MV had detrimental effects on diaphragmatic myofibrillar structures. These effects were reduced in HIF-1α-deficient mice.
  2. Endotoxemia + MV enhanced oxidative stress, upregulated mitochondrial DNA but downregulated Tfam. These effects were reduced in HIF-1α-deficient mice.
  3. Endotoxemia + MV increased Drp1 and reduced Mfn2; these effects were reduced in HIF-1α-deficient mice.
  4. Endotoxemia + MV downregulated PHD2 and upregulated HIF-1α; these effects were reduced in HIF-1α-deficient mice.
  5. Endotoxin-induced and concurrent MV-induced oxidative stress increase mitochondrial injury, autophagosomes, and light chain 3-II expression, and decrease P62 expression. These effects were reduced in HIF-1α-deficient mice.

The authors showed that MV and sepsis (e.g. endotoxins) exert synergistic effects on the development of severe diaphragm weakness, and this effects are reduced in HIF-1α knockout mice.

I compliment the authors for their impressive work: the work is clearly presented, the results are sound and novel, and potentially clinically significant, since diaphragmatic weakness is ubiquitarians in icu patients with sepsis.  

I have no major comments.

Minor comment:   

I would start the discussion summarizing the results of the study (Line 276 to 288). Thereafter I would continue the discussion as per Line 261 to 276.

Author Response

Minor comment:   

I would start the discussion summarizing the results of the study (Line 276 to 288). Thereafter I would continue the discussion as per Line 261 to 276.

Ans: Thanks for the reviewer’s suggestion, we have started the discussion summarizing the results of the study (from Line 276 to 288) and thereafter continue the discussion (from Line 261 to 276).

Reviewer 2 Report

This is an interesting paper and clearly of topic interest in view of the current COVID epidemic and the large number of patients who have to undergo MV due to hypoxia following sepsis. 

It is apparent from the results that there is at least an additive effect of MV and sepsis on mitochondrial oxidative stress that results in a deficit in the mechanical contractile strength of the muscle, probably due in the main to mitochondrial oxidative stress.  

The authors ascribe the main part of the damage to HIF1α since they note some improvements in mice with HIF1α (--).  Since recovery after MV and sepsis might be improved by suppression of HIF1α activity if might be worth testing whether HIF1α inhibitors or on the downstream effects of HIF1α activity e.g. LDH activity have any effect on VIDD. I note that there are reported effects of such inhibitors both in vitro (salinomycin), Dewangan et al, Biochemical Pharmacology 164 (2019) 326-335 ot with  CTM (chetomin), a small molecule inhibitor of HIF1α’s transcriptional activity see Hu et al (Circ Res. 2018;123:1066-1079. DOI: 10.1161/CIRCRESAHA.118.313249.

Obviously if these inhibitors prove to be useful in amelioration of VIDD this could have considerable clinical significance, albeit after careful clinical trials. 

Author Response

The authors ascribe the main part of the damage to HIF1α since they note some improvements in mice with HIF1α (--).  Since recovery after MV and sepsis might be improved by suppression of HIF1α activity if might be worth testing whether HIF1α inhibitors or on the downstream effects of HIF1α activity e.g. LDH activity have any effect on VIDD. I note that there are reported effects of such inhibitors both in vitro (salinomycin), Dewangan et al, Biochemical Pharmacology 164 (2019) 326-335 ot with  CTM (chetomin), a small molecule inhibitor of HIF1α’s transcriptional activity see Hu et al (Circ Res. 2018;123:1066-1079. DOI: 10.1161/CIRCRESAHA.118.313249.

Obviously if these inhibitors prove to be useful in amelioration of VIDD this could have considerable clinical significance, albeit after careful clinical trials. 

Ans: Thanks for the reviewer’s suggestion. We added "In an in vitro study of human umbilical vein endothelial cells, Dewangan et al. demonstrated that salinomycin, a monocarboxylic polyether ionophore obtained from Streptomyces albus, reduced angiogenesis of breast cancer cells progression through inhibiting HIF-1α/vascular endothelial growth factor signaling axis [51]. Hu et al. also reported that chetomin, a small molecule inhibitor of HIF-1α, repressed transcriptional activities of HIF-1α and its downstream target lactate dehydrogenase A [52]. This in turn resulted in an appropriate metabolic shift from aerobic glycolysis to oxidative phosphorylation and was accompanied by an increase in mitochondrial content and cellular ATP levels. Further investigation using these inhibitors is necessary to prove their effects in ameliorating VIDD" to Discussion on page 13, line 413-422.
